# The Prevalence and Impact of Innovative CSR Strategies in Manufacturing Enterprises in the Silesian Voivodeship: A Multifaceted Analysis of Benefits, Challenges, and Market Adaptability

Radosław Wolniak [1], Joanna Sadłowska-Wrzesińska [2], Ireneusz Miciuła [3,*], Henryk Wojtaszek [4], Maja Głuchowska-Wójcicka [5], Klaudia Skelnik [5], Roman Tylżanowski [6] and Żaneta Nejman [2]

[1]  Organization and Management Department, Silesian University of Technology, 41-800 Zabrze, Poland; radoslaw.wolniak@polsl.pl
[2]  Faculty of Management Engineering, Institute of Safety and Quality Engineering, Poznań University of Technology, 60-965 Poznań, Poland; joanna.sadlowska-wrzesinska@put.poznan.pl (J.S.-W.); zaneta.nejman@put.poznan.pl (Ż.N.)
[3]  Department of Sustainable Finance and Capital Markets, Institute of Economics and Finance, University of Szczecin, 70-453 Szczecin, Poland
[4]  Institute of Logistics, Faculty of Management and Command, War Studies University, 00-910 Warsaw, Poland; h.wojtaszek@pracownik.akademia.mil.pl
[5]  Department of Security Sciences, Faculty of Finance and Management, WSB Merito University, 80-266 Gdańsk, Poland; mgluchowska@wsb.gda.pl (M.G.-W.)
[6]  Department of Enterprise Management, Institute of Management, University of Szczecin, 70-453 Szczecin, Poland; roman.tylzanowski@usz.edu.pl
*  Correspondence: ireneusz.miciula@usz.edu.pl

**Abstract:** This article presents a comprehensive analysis of the occurrence of innovative solutions in accordance with CSR in various companies. It turns out that 71% of medium-sized enterprises (50–249 people) and 64% of smaller companies (10–49 people and up to 9 people) declare the introduction of innovative solutions in CSR strategies, which proves that innovations are not limited to large companies. A responsible approach to business can affect competitiveness and positive public perception. The implementation of a CSR strategy increases profits, improves image, increases customer loyalty, attracts media attention, and opens up new markets. The hindering factor is the additional cost of implementing CSR activities, insufficient knowledge and awareness of CSR, and unclear regulations and market conditions, as well as the need to adapt the organizational structure and culture. CSR strategies are important for the success of enterprises and society, and solving potential problems allows for effective introduction of responsible practices. In order to obtain the presented data, two research methods were used: an analysis of the international literature sources and a survey by manufacturing firms in the Silesian Voivodeship.

**Keywords:** CSR strategy; competitiveness; market demand; benefits; challenges; social innovation; economy

## 1. Introduction

Business management has been enhanced in recent years; it is necessary to introduce innovations that generate demand, accelerate the development of enterprises, and reduce product prices, meeting constantly growing human needs. Research in this area shows that innovations are closely related to the increase in the effectiveness and competitiveness of the company on the market [1]. The ability to create and absorb innovations is therefore a key challenge of the 21st century.

The analysis and assessment of innovativeness of enterprises are complex issues that often raise many uncertainties [2]. Innovation stems from many factors, both internal

and external to the company [3]. These factors include the micro-environment, environmental diversity, scale of operation, organizational structure, business strategies, and entrepreneurial ambitions, as well as creativity and willingness to take risks related to innovation. A crucial part is also attributed to studies and the contemporariness of the adopted strategies as prerequisites for the triumph of innovation. Yet, the character of innovative procedures is intricately multifaceted, posing challenges in clearly outlining the conditions influencing businesses' inclination towards devising such methods [4]. This is why so much research is being carried out in this area.

Our incentive to delve into this subject is further driven by the present growth of the examined businesses within a particular nation and distinct geographical region. Reviewing numerous studies enables us to pinpoint the common prerequisites for the inventive endeavors of fundamental economic entities, namely, businesses. Key elements contributing to the success of innovation include independent thought, the capacity to critically evaluate the real world, and the willingness to embrace risks. It is also worth noting that not all alterations qualify as innovations; they might just be ineffective shifts or replicas of pre-existing methods. The aim of the article is a multiaspect analysis of the benefits resulting from innovative CSR strategies in manufacturing enterprises in the Silesian Voivodeship, as well as current challenges in this field and the adaptability of the presented market.

## 2. Literature Review

Innovation is the transformation of ideas and creativity into practical and valuable products, services, or technologies [5]. It demands investigative, structural, fiscal, and promotional efforts. The significance of innovation fluctuates throughout the different phases of a business's lifespan. A business centered on innovation possesses the capability to convert unique ideas into tangible outcomes, foster inventiveness within its workforce, and effectively vie in the marketplace [6]. Being innovative denotes the capacity to birth something novel, essentially a method to actualize creative thoughts and bring them to fruition. It is an act of launching new, or substantially revamped, goods, services, or operational processes into the market, as well as embarking on fresh marketing endeavors and ushering in organizational shifts that modify the firm's modus operandi and affiliations with its external surroundings [7]. An inventive approach is recognized as an innovation only once it is integrated into the business's procedures. Innovation propels a surge in productivity, which subsequently boosts the firm's competitive edge in the market place [8].

Technological advancement arises when a novel, or enhanced, product reaches the marketplace, or when an updated or superior production method is employed. The firm might obtain fresh technology from external sources, or update its existing operational techniques. Explorations into new uses for current products are also undertaken. Cutting-edge technology serves as a pivotal competitive asset, facilitating the launch of novel and better products into the market and refining promotional endeavors [9].

The cyclical theory of economic progression owes its evolution not only to Joseph Schumpeter but also to the Russian economist N.D. Kondratiev (1935) [5–7,9], who proposed the notion of elongated patterns and sequences of industrial transformation in the 20th century's early years. Kondratijewa chronicled the forward motion of industry in terms of four extended patterns associated with shifts in the technoeconomic framework. Per his doctrine, transformations in the fiscal structure, from the late 18th century onward, were instigated by groundbreaking innovations. At present, numerous scholars revisit this notion, discerning not just economic but spatial facets in it.

The investigative team intrigued by N.D. Kondratieva's long-wave theory comprises personalities like N. Rosenberg and C. Frischtak (1984), Ch. Freeman and C. Perez (1988), A. Grübler and H. Nowotny (1990), R. Hayter (1997), P. Dicken (1998), T. Stryjakiewicz (1999), and P. Haggett (2001) [6–8,10]. These academicians discern in Kondratieva's theory both economic and spatial regularities. Of particular import is the origin locale of a specific revolutionary innovation [10]. Another notion expanding upon Schumpeter's ideas of economic cycle periodicity is G. Mensch's theory (1979) [11]. Mensch explored the

framework of cyclical structural shifts, presuming the economy advances through sporadic innovation sparks, crafting serial patterns akin to the letter "S". Much like Kondratijewa, Mensch also highlighted the collective emergence of radical innovations, marking a notable conceptual addition to innovation studies.

Among doctrines addressing economic growth, Michael Porter's cluster concept is notable. Clusters represent conglomerates of companies and affiliated establishments that specialize in a distinct domain, bound by shared characteristics and reciprocal enhancements [12]. Such clusters can emerge on diverse geographic magnitudes, spanning a single metropolis, province, nation, or a coalition of adjacent nations. Clusters proffer numerous advantages to investors and can exploit synergistic endeavors.

Within locational theory, innovation can also be evaluated in tandem with the idea of spatial self-structuring, illustrating a multifaceted economic system. This doctrine portrays the transitions complex systems undergo, presumed to be open, nonlinear, and distant from equilibrium. Analogous to physics theory, economics examines shifts from one state to another. A disparity in physics might arise from variety, while in economics, it might precipitate divergent operational modalities in zones like cities or provinces, which might have varied employment or investment structures. Nonlinearity might equate to scale economies, and disturbances to the ramifications of new technology introductions sparking innovation and expansion [13].

In societal contexts, innovations are not confined to technical and technological creations but also span organizational and promotional strategies, and the foundation of societal entities bolstering innovative activities within organizations. Peter Drucker perceived innovation as an essential instrument assisting entrepreneurs across various sectors. Via innovations, businesses can tackle emerging fiscal challenges. The Oslo Manual's innovation definition encompasses the introduction of a novel, or markedly enhanced, product, service, method, marketing approach, or workflow to the marketplace [14]. Drucker enumerates seven innovation avenues, four relating to businesses' internal endeavors: sudden market triumph or debacle, when reality conflicts with perceptions, innovation driven by process necessities, and alterations in market or industry structures [15,16].

To catalyze innovative endeavors, entities can instigate organizational shifts, eliminate obstructions, adopt adaptable stances, and leverage tangible incentives [17]. Personnel should readily access innovation-essential resources, like procuring novel technologies or funding, and should harbor a risk-taking mindset. Cross-functional teams and task groups bolstering organizational innovation are pivotal. A robust work ethos promotes trials, celebrates accomplishments, and mitigates failure risks [18,19]. Innovations delivering fresh consumer value often materialize as new products, technologies, concepts, methodologies, and systems, primarily yielding client advantages. Consequently, this bolsters client fidelity and the enterprise's robust profitability [20,21]. With the service sector's ascension, the innovation spectrum has broadened extensively, transcending the technical realm. Innovations can be conceived as products, services, or ideas perceived as fresh by diverse stakeholders [22,23]. To maintain their market edge, enterprises must incessantly fortify their stance, outperforming rivals in unearthing inventive and elusive solutions. Only firms that perennially cultivate innovation will endure and flourish in the market. Key is the stewardship of in-house research and developmental activities [24]. However, research and development investments in Poland remain inadequate. Limited collaborations with academicians and academic institutions also stymie the tangible deployment of inventive solutions [25].

The ability of businesses to compete is influenced by a multitude of elements, both from external and internal environments. External influences encompass governmental strategies, organizational structures that aid businesses, tangible and intangible resources, as well as interactions with business partners and adversaries. Internally, the management approach, operational funds, adoption of contemporary methods and technologies, and the caliber of produced goods play pivotal roles. Various categorizations exist concerning the prerequisites for Innovation within companies [1,26].

To enhance the innovation capabilities of personnel, it is crucial to emphasize the advantages of engaging in educational opportunities and furthering their professional development. It is essential to disseminate knowledge and motivate employees to continuously refresh their understanding. It is important that creative workers feel secure in employment, so that they are not afraid of sanctions for making mistakes when introducing innovations. This is how employees emerge who become promoters of progress. Avant-garde innovations are characterized by self-confidence, perseverance, energy, and willingness to take risks. Effective management is conducive to innovation, creating a favorable climate to inspire and promote the company's vision and development strategy. This, in turn, motivates employees to undertake initiatives worthy of recognition. Freedom of action helps employees adapt to innovative solutions. The innovation process is not limited to triggering initiatives and consolidating activities, but also includes recognizing the stress associated with it and identifying ways to mitigate it.

Both emerging and established businesses need to distinctly identify the market they aim to engage in, and the approach to sustain a competitive edge. Being competitive is evident in delivering superior-quality products, the efficacy of marketing endeavors, all rooted in innovative principles. An enterprise can carve out a competitive lead by amplifying its manufacturing capabilities, possessing unique assets, particularly its workforce, and guaranteeing client contentment [15,27]. Pioneering initiatives play a pivotal role in market competition.

The choice of an appropriate strategy by an enterprise determines its ability to operate in a given industrial sector and affects the chances of survival and maximum profits. However, strategies focused solely on maximizing profits are not always beneficial to society as a whole [28,29]. Achieving a competitive advantage depends not only on important factors of production, but increasingly on the company's development strategy [30,31]. Determining what the company wants to excel in and how to achieve a competitive advantage is a key starting point. The motives for acting for the environment are usually addressing their self-interest by ensuring access to resources in the future or meeting the expectations of stakeholders representing environmental needs [32]. Activities for customers or suppliers under the CSR directly impact the company's cooperation with these stakeholders [33]. CSR is cultivating a positive attitude of customers, partners, stakeholders, public opinion, and the local community.

The company's organizational framework should be strategically crafted to promote competitive behavior, and the approach to competition should be deliberate and efficient. Recognizing the factors influencing demand at the local, regional, national, and global levels is also vital. The presence of interconnected industries and services within a specific region aids in boosting innovation and competitive edge.

Competition in the country is conducive to the development of innovation and product quality, and the experience gained is an added value in competing on the international arena. International trade has a positive impact on the development of innovation, stimulates higher quality products, lowers production costs, and increases productivity. However, import restrictions, administrative fees, and border controls may limit the reaction time to changing conditions that favor or hinder innovative ventures [34,35]. For an innovation-oriented company, it is important to search for new solutions, business concepts, and ideas. Creativity, the ability to take advantage of opportunities, take risks, and adapt to the environment is of key importance to the innovation process. Competition on the market, continuous improvement of product quality, and flexibility in adapting to the changing needs and preferences of customers are the key elements stimulating innovation [36].

The innovativeness of the country and enterprises depends on many factors that determine the scope and level of economic activity. The key factors influencing the innovativeness and competitiveness of enterprises are the capacity to devise, originate, and apply innovations; proficiency in assimilating novel approaches; structural skills to bolster the competitive stance, rooted in technical and administrative capacity; expertise in advancing innovation and diversifying products, drawing from technical, technological, and monetary

assets; and the innovation prowess stemming from technical and technological capabilities, namely, the allure and novelty of the technologies owned [37–42]. Enterprises striving to maintain a sustainable competitive advantage must constantly improve their operations and take actions to improve them. Focusing on innovative activities is a key element in achieving the development goals of the company, and it also favors the emergence of new economic areas and sectors. Contemporary enterprises should be organizations focused on innovation [43–45]. The word "innovative" pertains to businesses that participate in thorough research and development endeavors or capitalize on the outcomes of efforts conducted beyond their own premises.

Such enterprises invest significant financial resources in this activity. They systematically introduce new scientific and technical solutions, and their production or offered services are based on a large number of innovative elements. Creating innovations and their implementation in production, work organization, and the market is an integral part of their activity [24,29,46–48].

The concept of responsibility has been thoroughly described in the literature on the subject. It is worth noting that responsibility can be understood as a value, as well as an incentive for employees to engage in community activities. The concept of responsibility evolves over time, partly reflecting changing social norms and values [49–51]. Responsibility is considered an ethical norm, which is related to the readiness to accept the consequences of one's decisions, both positive and negative. It can be both forced and voluntary [52–56]. If it is enforced, it may result from the law or pressure from organized groups. However, in the context of the presented article, voluntary responsibility is particularly important, which means the choice to undertake socially responsible activities [57,58].

In the literature, a lot of attention is devoted to corporate responsibility in three areas: economic, social, and ecological, which includes [59–66]:

- Maintaining fair competition, building relationships with customers based on reliability and care for honesty towards suppliers (economic aspect).
- Involvement in social and cultural life, respect for law, tradition, and protection of cultural heritage; special emphasis is placed on supporting health-promoting activities (social aspect).
- Compliance with the principles of sustainable development, protection of natural resources, reduction of pollution, and activities for the protection of the natural environment (ecological aspect) [67–69].

Among the various definitions of corporate social responsibility, there are also those that emphasize the importance of security issues. The Industry Canada Association lists the areas covered by CSR: health and safety, human rights, human resources management, corporate governance, business ethics, community development, consumer rights, and protection of the workforce. In addition, it addresses issues related to relations with suppliers and takes into account the rights of stakeholders and environmental protection. The ISO 26000 standard, commonly known in the context of CSR, emphasizes the responsibility of the organization towards society and the natural environment in decisions and activities related to products, services, and processes [70–73]. This standard emphasizes transparent and ethical behavior that contributes to sustainable development, health, and welfare of society, while taking into account the expectations of stakeholders [74–76]. The implementation of the stakeholder approach, which is crucial in the context of CSR, may differ depending on the type of corporate governance. Currently, the subject of CSR is also being extended to the activities of nonprofit organizations that have mainly social goals. The need to implement CSR by the armed forces is also indicated [77–79].

Currently, in the face of the growing interest in the analysis of crisis situations, including those of a noneconomic nature, the approach indicating the role of CSR in strengthening the security of the organization is beginning to gain importance [80–83]. Therefore, the thesis can be justified that the social and ecological responsibility of the organization, as part of continuous cooperation with stakeholders, should take into account the need to think about building the organization's resistance to potential threats. This resilience can

be related to the culture of the organization, which takes into account the values related to CSR, i.e., social and ecological responsibility. In the literature related to this topic is the so-called security pyramid, according to which, with a high level of safety culture, the organization should first focus on ensuring ecological and social security. On this basis, economic and political security should be built, and then other areas of security, such as public or military [83–85].

## 3. Materials and Methods

The purpose of the investigation was to perform an assessment regarding the social and eco-friendly responsibility of innovation endeavors by manufacturing firms in the Silesian Voivodeship. Central to this was identifying the presence of sustainable innovations (namely, innovations with a socially and environmentally friendly orientation aligning with the sustainable development framework). These included small-, medium-, and large-scale businesses.

The inquiry validated several common elements that impact the innovative capacity of companies on a worldwide level. It was discerned that insufficient financial resources for growth, antiquated technology, and the absence of skilled workers are the primary reasons for the infrequent adoption of innovations in the majority of the evaluated manufacturing firms. Notably, it was revealed that the primary hindrances are not contingent upon the company's age. This research yields multiple insights about businesses across different stages of national progression and external dynamics that affect commercial operations. For instance, in the context of the Silesian Voivodship, it was observed that a firm's success is tethered to the competitive edge derived from the nature of innovations implemented. Companies that introduced process innovations and operated on the market for a long time achieved better results. This finding contributes to a new analytical contribution to our understanding of the drivers of innovation. CAWI (Computer-Assisted Web Interview) is a technique for collecting information in quantitative market and public opinion research, in which the respondent is asked to complete a survey in electronic form. Online research (CAWI) is one of the forms of quantitative measurements, which involves conducting research with using survey questionnaires provided electronically. Thanks to the online survey mode, online research can be carried out on large groups of respondents, while ensuring the anonymity of the respondents and the possibility of conducting many independent measurements in parallel, and significantly reducing the time and cost of conducting the research.

The group of respondents comprised owners or managers of manufacturing companies and their employees. The research focused on manufacturing companies in the Silesian Voivodeship. The study covered the Silesian Voivodeship region, and the research period extended from 2020 to 2023. A total of 310 manufacturing companies were examined during this period. The primary constraint of the research sample was the budgetary and temporal boundaries of the endeavor. Statistical bureaus, municipal departments, and countryside administrations lacked data concerning the current state of operational enterprises. The absence of data was ascribed to multiple causes:

- Inconsistency in providing information by entrepreneurs, for example, due to business cessation.
- Changes in formats for reporting economic entities.
- Intentional deceptive behavior by entrepreneurs, such as operating in the informal economy.

The data from the Central Statistical Office were considered the most reliable source of information in the Silesian Voivodeship. Population data were generated from the Central Statistical Office and updated based on data from the Central Statistical Office in Katowice. The group of manufacturing companies included companies classified by the Central Statistical Office in Section C of the Classification of Economic Activities as "Manufacturing", including:

(1)    Division 10: Manufacture of foodstuffs;

(2)　　Division 11: Production of beverages;
(3)　　Division 12: Manufacture of tobacco products;
(4)　　Division 13: Manufacture of textiles;
(5)　　Division 14: Manufacture of clothing;
(6)　　Division 15: Manufacture of leather and related goods;
(7)　　Division 16: Manufacture of wood and products of wood and cork, except furniture; manufacture of articles of straw and plaiting materials;
(8)　　Division 17: Manufacture of paper and paper products;
(9)　　Division 20: Manufacture of chemicals and chemical products;
(10)　Division 21: Manufacture of basic pharmaceutical products, drugs, and pharmaceutical preparations;
(11)　Division 22: Manufacture of rubber and plastic products;
(12)　Division 23: Manufacture of other nonmetallic mineral products;
(13)　Division 24: Production of basic metals;
(14)　Division 25: Manufacture of fabricated metal products, excluding machinery and equipment;
(15)　Division 26: Manufacture of computers, electronic and optical products;
(16)　Division 27: Manufacture of electrical equipment;
(17)　Division 28: Manufacture of machinery and equipment n.e.c.;
(18)　Division 29: Manufacture of motor vehicles, trailers, and semi-trailers, except motorcycles;
(19)　Division 30: Manufacture of other transport equipment;
(20)　Division 31: Manufacture of furniture;
(21)　Division 32: Other production.

For preliminary research, 10% of the companies that met the time and spatial criteria were selected. A random number tool was utilized to select which firms from the roster would be part of the study group. Invitations to partake in the research were dispatched to the chosen firms. Affirmative replies were obtained from 310 firms. Given the scope of the sample (encompassing both firms and their employees), alongside the significant expenses and extended duration of the study, there was no increase in the sample size. To gauge employee innovativeness, 2 staff members were chosen from each micro-business, 5 from minor firms, 10 from medium firms, and 20 from major firms, culminating in an overall sample of 911 staff members.

## 4. Results

The research included 310 proprietors and administrators of production businesses functioning in the Silesian Voivodeship. These businesses were chosen at random from the entire set of production companies, leading to a varied sample with differences in both operational profiles and durations. The resulting sample was arbitrary and validated via sequential testing.

Figure 1 below shows the structure of the surveyed manufacturing companies based on the number of employees. The analysis of employment in these companies is a key factor in understanding the diversity and scale of the manufacturing sector. The presented data were collected as part of a study conducted in the Silesian Voivodeship. The value of Figure 1 lies in showing the percentage share of enterprises in various employment categories, such as micro, small, medium, and large enterprises.

Figure 1 shows the structure of the surveyed production companies, broken down by the size of employment. The data source is the authors' own elaboration. According to the presented distribution, it can be seen that most of the surveyed manufacturing companies (68%) fall within the range of 50 to 249 employees. This means that the majority of the production sector in the study area is represented by medium-sized enterprises. The second-largest group includes 24% of the surveyed companies that employ from 10 to 49 employees. These are mainly small and medium-sized enterprises. The smallest group are companies employing up to 9 employees, which constitute 8% of the surveyed enterprises. These are mainly micro-enterprises or small companies that are characterized by a smaller scope of

activity. Interpreting this figure, it can be seen that most of the surveyed manufacturing enterprises belong to the category of medium-sized enterprises, which may suggest that this segment of the sector is of particular importance for the economy of the surveyed area. Enterprises in this category may have greater resources, production capacity, and market impact compared to smaller firms.

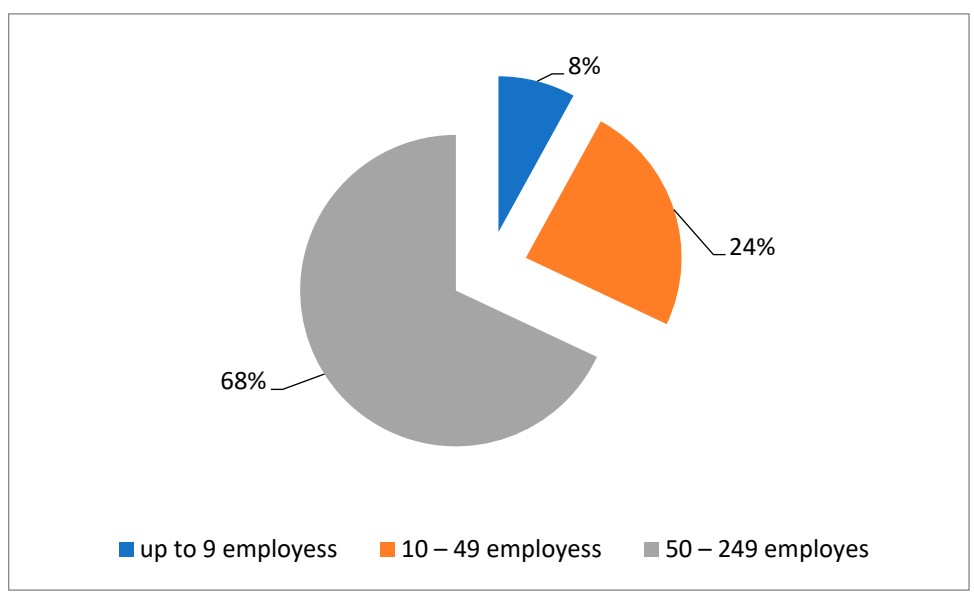

**Figure 1.** The structure of the surveyed production enterprises by the size of employment. Source: Authors' study based on the obtained results.

The study of the structure of employment in companies provides valuable information for economic analysis, strategic planning, and economic policy [84]. Such data can help to understand the dynamics and characteristics of the manufacturing sector, as well as identify areas that need support or further development.

Enterprises employing up to nine people have an average age of 19 years. This means that the smallest enterprises have been in existence for 19 years on average. Companies with 10 to 49 employees also have an average age of 19. Therefore, companies of this size have also been in existence for an average of 19 years. Enterprises that employ between 50 and 249 people have a slightly longer average age of 24. This means that medium-sized enterprises have been in existence for an average of 24 years. Analyzing these results, it can be seen that enterprises in the researched area have a different length of existence depending on the size of employment. Smaller enterprises (employing up to 9 and from 10 to 49 persons) have a similar average age of 19 years, while medium-sized enterprises (employing from 50 to 249 persons) exist a bit longer, on average for 24 years. It is worth noting that the age of the enterprise may affect its experience, financial stability, and ability to survive on the market. Older enterprises may have richer experience and a better developed network of customers and business partners, which can contribute to their success. In contrast, younger companies may be more innovative and dynamic, but at the same time more exposed to the risks associated with the start-up phase.

Figure 2 shows the structure of the surveyed enterprises according to the scope of their activity. Enterprises were divided into three categories: regional, national, and international.

The results show that 18% of the surveyed enterprises operate at the regional level, i.e., they offer their services or products in a limited geographical area, usually in one specific region. On the other hand, 48% of enterprises operate at the national level, which means that their services or products are available throughout the country. These types of companies have a national scope and operate in many areas or cities in the country. The largest group are international companies, which represent 34% of the respondents.

These companies operate on an international scale, which means that they have branches, subsidiaries, or operate in many countries around the world.

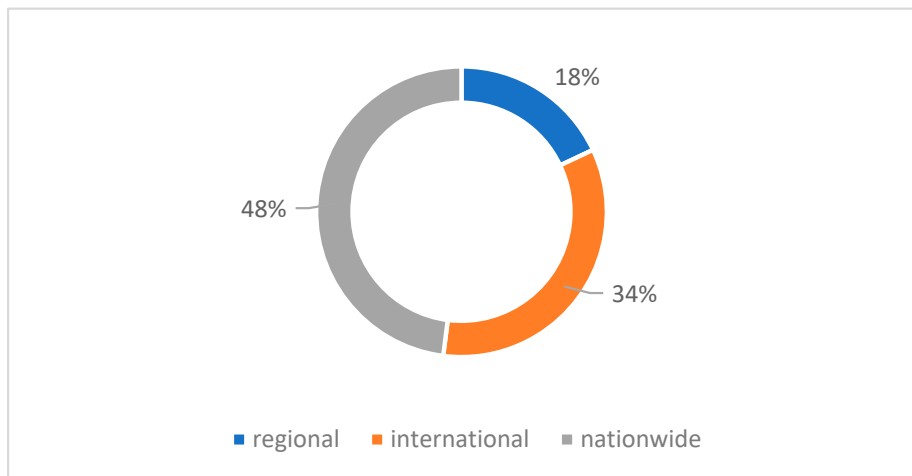

**Figure 2.** The structure of the surveyed enterprises according to the scope of activity. Source: Authors' study based on the obtained results.

These results suggest that most of the surveyed companies are national or international in nature, indicating their growing reach and potential impact on domestic and global markets. Nationwide and international companies often face more complex governance, competition, and social responsibility challenges, which may require more comprehensive strategies and approaches. On the other hand, regional enterprises, concentrated in a smaller area, may be more focused on the needs of local communities and adapted to regional conditions.

Figure 3 presents the findings of a research study regarding the familiarity with the concept of CSR based on the size of the enterprises. The results were categorized into three groups: 50–249 companies, 10–49 companies, and fewer than 10 companies.

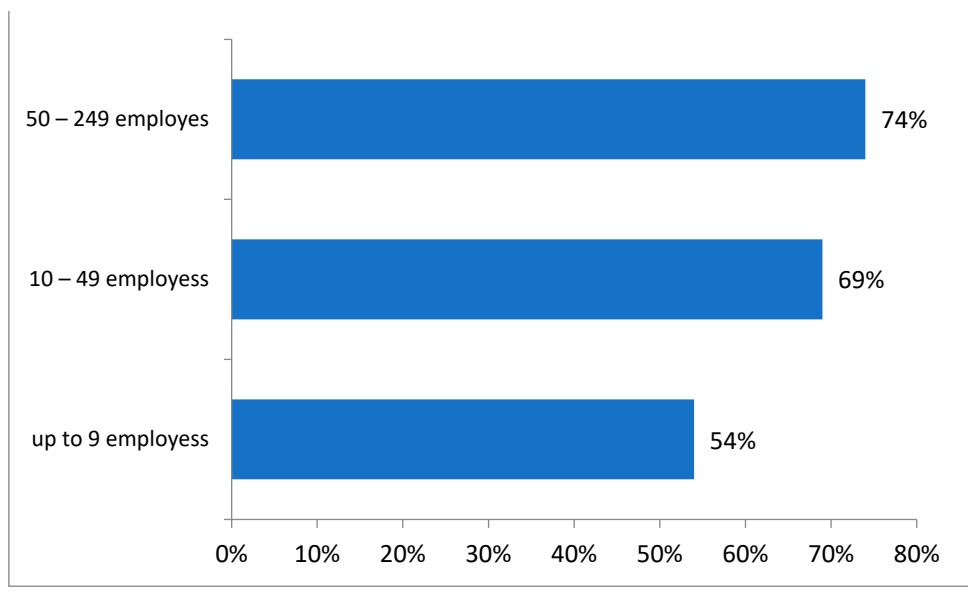

**Figure 3.** Knowledge of the CSR concept by company size. Source: Authors' study based on the obtained results.

Among companies categorized as 50–249, an impressive 74% of respondents reported having a solid understanding of the CSR concept. This indicates that a significant majority

of participants in this group comprehend the essence of CSR. In companies classified as 10–49, 69% of respondents claimed familiarity with the CSR concept. While slightly lower than the percentage in larger enterprises, this group still demonstrates a commendable level of awareness regarding CSR. In the smallest enterprises (fewer than 10 companies), 54% of respondents stated that they are acquainted with the concept of CSR. Although this figure is relatively lower than the other two groups, it is noteworthy that over half of the participants in this category understand what CSR entails. Overall, the results suggest that familiarity with the CSR concept is relatively widespread among the surveyed enterprises, regardless of their size. However, larger companies appear to exhibit slightly higher levels of awareness compared to their smaller counterparts. There is potential for further education and awareness-raising efforts concerning CSR, especially among smaller companies, to foster more responsible business practices and generate a positive social and environmental impact.

Figure 4 shows the assessment of the occurrence of innovative solutions based on the CSR strategy depending on the size of the enterprise. The results were divided into three categories of companies: 50–249 companies, 10–49 companies, and up to 9 companies.

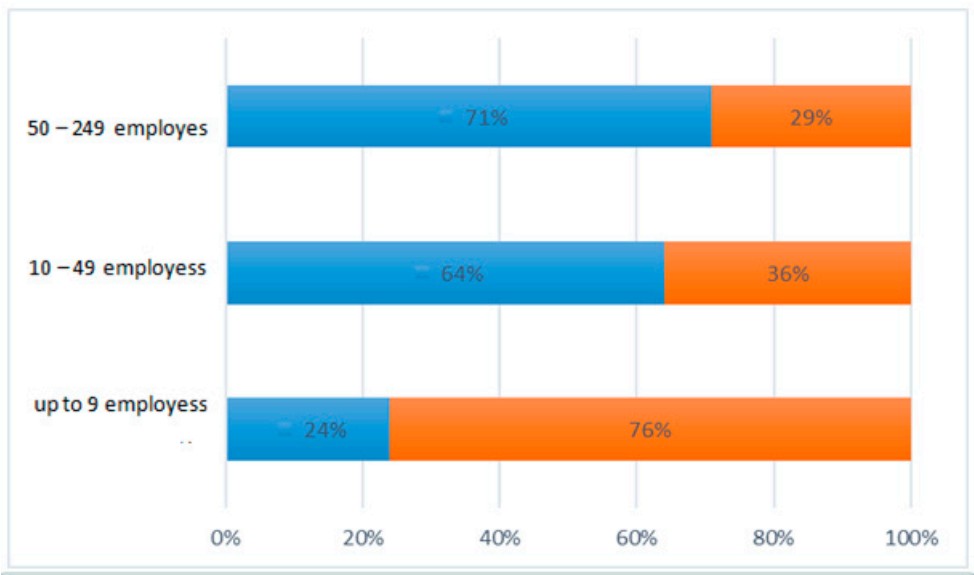

**Figure 4.** Assessment of the occurrence of innovative solutions based on the CSR strategy. Source: Authors' study based on the obtained results.

Enterprises from the 50–249 category have 71% of the assessed occurrence of innovative solutions based on the CSR strategy. This means that the majority (71%) of medium-sized enterprises have innovative solutions implemented in their CSR strategy. Enterprises from the category of 10–49 companies have 64% of the assessed occurrence of innovative solutions based on the CSR strategy. This means that 64% of these enterprises declare that they have innovative solutions related to the implementation of CSR strategies. In the group of companies with up to 9 employees, the percentage declaring this type of solutions is only 24%. Analyzing these results, it can be seen that regardless of the size of the company, a relatively similar percentage of companies declare the presence of innovative solutions in their CSR strategies. Medium-sized companies show a slightly higher percentage (71%), while smaller companies (10–49 companies and up to 9 companies) have a slightly lower result, but are still at the level of 64%. This suggests that innovative solutions related to CSR are not limited to large enterprises. Small and medium-sized companies also see the importance of innovation in their activities related to CSR. This approach can contribute to increasing competitiveness, positive impact on society and the environment, and creating a positive image of the company.

Figure 5 shows the assessment of the occurrence of innovative solutions in individual areas, taking into account compliance with CSR depending on the size of enterprises. The results were divided into three categories of companies: 50–249 companies, 10–49 companies, and up to 9 companies.

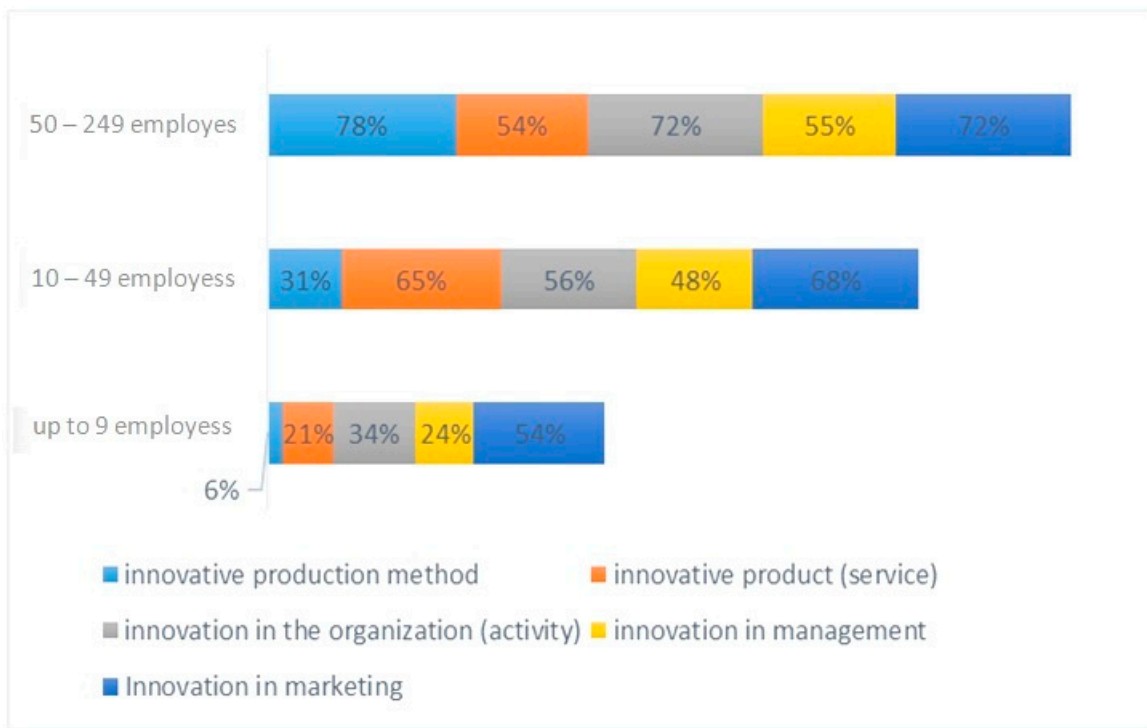

**Figure 5.** Evaluation of the occurrence of innovative solutions in individual areas based on compliance with CSR. Source: Authors' study based on the obtained results.

Enterprises from the 50–249 category present a higher level of innovative solutions based on the CSR strategy compared to the other two groups. In the area of "innovative production methods" the percentage is 78%, and in "innovation in the organization (activity)" and "innovation in marketing" it is 72%. In the case of "innovative products", the percentage is lower, but still significant, at 54%. In "innovation in management" the percentage is 55%.

Companies in the 10–49 category of companies present variable results in various areas. In "innovative production methods" the percentage is relatively low at 31%. In "innovation in organization (business)" and "innovation in marketing", the percentages are 56% and 68%, respectively, which is quite competitive. In "innovative products" the percentage is 65%, and in "innovation in management" it is 48%. Enterprises in the category of up to nine companies have the lowest level of innovative solutions in all areas. In "innovative production methods" the percentage is very low and amounts to only 6%. In "innovation in organization (activity)" and "innovation in marketing" the percentages are 34% and 54%, respectively, which is lower compared to larger companies. In "innovative products" the percentage is the highest among other fields and amounts to 21%, while in "innovation in management" the percentage is 24%. Analyzing these results, it can be seen that larger companies (50–249 companies) have a much higher level of innovation in all the researched areas compared to smaller companies (10–49 companies and up to 9 companies). Especially in the area of "innovative production methods" and "innovation in the organization (activity)", the differences are significant. Smaller companies show a lower propensity to introduce innovative solutions in their CSR strategies, which may result from differences in the resources and capabilities of the surveyed groups. These results suggest that smaller companies have the potential to improve their innovation and

CSR activities, which can contribute to increased competitiveness and positive social and environmental impact. In addition, the implementation of innovative solutions in the area of CSR can bring benefits to both the company and its stakeholders.

Figure 6 shows the assessment of the occurrence of different types of innovative solutions in accordance with the concept of CSR depending on the size of enterprises. The results were divided into three categories of companies: 50–249 companies, 10–49 companies, and up to 9 companies.

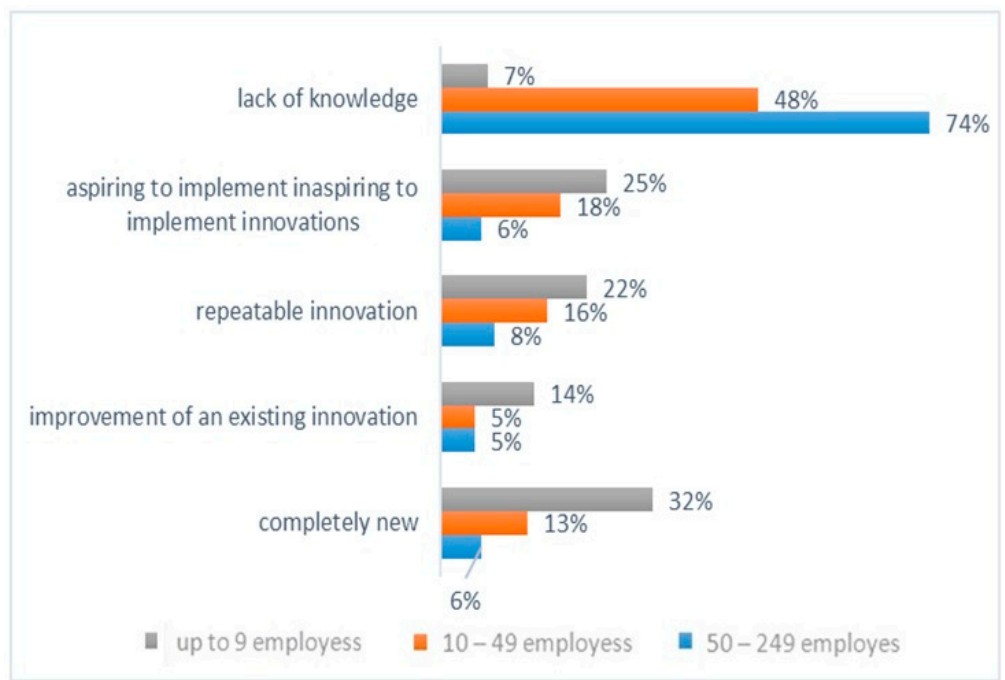

**Figure 6.** Assessment of the occurrence of types of innovative solutions in accordance with the CSR concept. Source: Authors' study based on the obtained results.

Enterprises from the 50–249 category are characterized by a diverse presence of various types of innovative solutions in accordance with the CSR concept. The largest group are "completely new" (completely new innovations)—32% of respondents declare the introduction of such innovations. Then we have "aspiring to implement innovation" (striving to implement innovation and lack of such striving)—25%. The next place is occupied by "repeatable innovation"—22%. "Improvement of an existing innovation" includes 14% of respondents. The smallest percentage is "lack of knowledge"—7%.

Enterprises from the 10–49 category also show differences in the occurrence of individual types of innovative solutions in accordance with the CSR concept. The largest group is "lack of knowledge"—as many as 48% of respondents declare a lack of knowledge on this subject. In second place is "aspiring to implement innovation" (striving to implement innovation and lack of such striving)—18%. The next places are occupied by "repeatable innovation" (repeatable innovations)—16% and "completely new" (completely new innovations)—13%. The smallest percentage, only 5% of the respondents, is the "improvement of an existing innovation" group. Enterprises in the category of up to nine companies are characterized by the most homogeneous distribution among all categories. The "lack knowledge" group has the highest percentage—74% of respondents from this group declare a lack of knowledge about innovations consistent with the concept of CSR. Then we have "completely new" (completely new innovations)—6%. "Repeatable innovation" and "improvement of an existing innovation" have 8% of respondents each. The smallest percentage, 6%, is the group "aspiring to implement innovation" (striving to implement innovation and lack of such striving).

Analyzing these results, it can be seen that the largest enterprises (50–249 companies) are characterized by the greatest diversity in terms of types of innovative solutions compliant with the concept of CSR. A large percentage of respondents in this group declare the introduction of completely new innovations and the desire to implement innovations. Smaller enterprises, especially those in the category of up to nine companies, tend to show a greater lack of knowledge about innovative solutions compliant with CSR. Introducing educational activities and support for these companies can help raise awareness of innovation and its social and environmental benefits. This analysis aims to understand how the implementation of the CSR strategy can affect the competitiveness of the company and the demand for its products and services. The key benefits that may result from the company's involvement in activities in accordance with the principles of CSR and the challenges that may arise during this process will be presented.

Figure 7 presents an analysis of the benefits of implementing the CSR strategy in the context of the impact on competition and market demand. The CSR strategy assumes undertaking responsible social and environmental actions by enterprises, which contributes to positive effects both for the company itself and for the business environment.

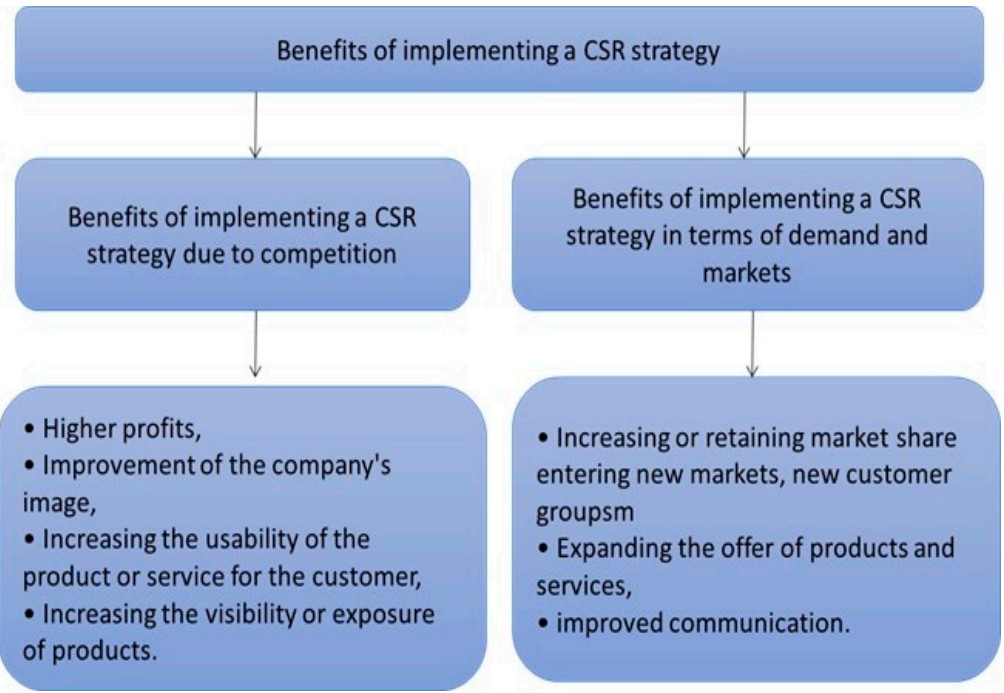

**Figure 7.** Benefits of implementing the CSR strategy—impact on competition and demand. Source: Authors' study based on the obtained results.

The implementation of a CSR strategy can bring a number of benefits both in terms of competition and market demand. Here is an overview of these benefits:

1. Benefits of implementing a CSR strategy due to competition:
   - Higher profits: Companies implementing a CSR strategy often achieve higher profits. This is due to the fact that socially responsible companies can enjoy a better image, greater customer loyalty, and more engaged employees, which translates into increased operational efficiency and increased profits.
   - Improvement of the company's image: Implementation of the CSR strategy allows the company to build a positive image among customers, investors, business partners, and society. Customers often prefer companies that are actively involved in activities for the benefit of society and the environment, which translates into greater trust and loyalty.

- Increasing the usefulness of a product or service for the customer: Implementing CSR can lead to an improvement in the quality of products and services and the inclusion of social and environmental responsibility throughout the supply chain. This can make the products or services offered more attractive to customers, which translates into increased customer loyalty and satisfaction.
- Increasing the visibility or exposure of products: A company that consistently implements a CSR strategy can gain more media, public, and customer attention. Positive actions and social involvement can attract media attention, which allows an increase in the exposure and recognition of products or services.

2. Benefits of implementing a CSR strategy in terms of demand and markets:

- Increasing or maintaining market share: The introduction of a CSR strategy can help a company to increase its competitiveness in the market, leading to an increase in market share. Customers often prefer companies that care about society and the environment, which may translate into greater interest and demand for the company's products.
- Entering new markets, new customer groups: A company that actively engages in CSR activities can reach new customer groups and open up to new markets. The growing number of customers who prefer socially responsible companies creates opportunities for expansion into new business areas.
- Broadening the offer of products and services: Implementing a CSR strategy can encourage a company to expand its offer with more ecological, responsible, and socially beneficial products and services. This can attract new customers and increase the variety of the company's offer.

3. Improving communication: Implementing a CSR strategy can help a company communicate better with customers, investors, the public, and other stakeholders. Positive actions related to social responsibility can attract media attention, which facilitates communication and promotion of the company and its products.

In conclusion, the implementation of the CSR strategy brings numerous benefits to the company both in terms of competition and market demand. Improving the image, increasing customer loyalty, greater operational efficiency, and diversity of the offer are just some of the benefits that can be achieved thanks to a responsible approach to business. In connection with sustainable development, the results of the study suggest that financial support, modernization of technology, and investing in the development of employees' competences can contribute to increasing the innovativeness of enterprises. The pursuit of process innovation can positively affect the competitiveness and long-term stability of enterprises, which is an important aspect of sustainable development [85]. Striving for a balance between economic growth and care for the natural environment and society can provide a sustainable competitive advantage in the global economy.

Figure 8 shows the benefits of implementing the CSR strategy in the context of increasing the possibilities of production and delivery of products and services. This model shows the positive effects that a company can achieve by taking actions in accordance with the CSR principles in the areas of production and delivery.

Benefits of implementing a CSR strategy towards increasing production capacity (production):

- Reducing operating costs related to the provision of services: CSR activities, such as sustainable management of resources or reducing waste, can contribute to the reduction of operating costs related to the production and provision of services.
- Increasing the efficiency or speed of delivering products and services: Introducing innovations and optimizing production processes in the spirit of CSR can accelerate the delivery of products or services to customers.
- Improving the quality of products and services: Engaging in a CSR strategy may result in improving the quality of products and services offered, which in turn contributes to greater customer satisfaction.

- Reducing unit labor costs—savings: CSR activities, such as investing in employee training or improving working conditions, can result in a reduction in unit labor costs.
- Achieving sector technical standards: By implementing a CSR strategy, the company can meet the required technical and quality standards applicable in a given industry, which facilitates cooperation with other entities.

Benefits of implementing a CSR strategy towards increasing delivery capabilities (delivery):

- Reducing the consumption of materials and energy: CSR activities, such as the use of energy-saving technologies or reducing the consumption of raw materials, can contribute to reducing the negative impact on the natural environment.
- Increasing production or service capacity: Engaging in a CSR strategy can contribute to increasing production or service capacity, which enables better meeting of the growing demand in the market.
- Shortening the production cycle: Implementing a CSR strategy can speed up production and delivery processes, shortening the time between ordering and delivery of products or services.
- Improving IT capacity: Investing in modern IT solutions can improve production and delivery management.
- Reducing product design costs: CSR activities, such as designing products with sustainable use of resources in mind, can reduce the cost of designing new products.

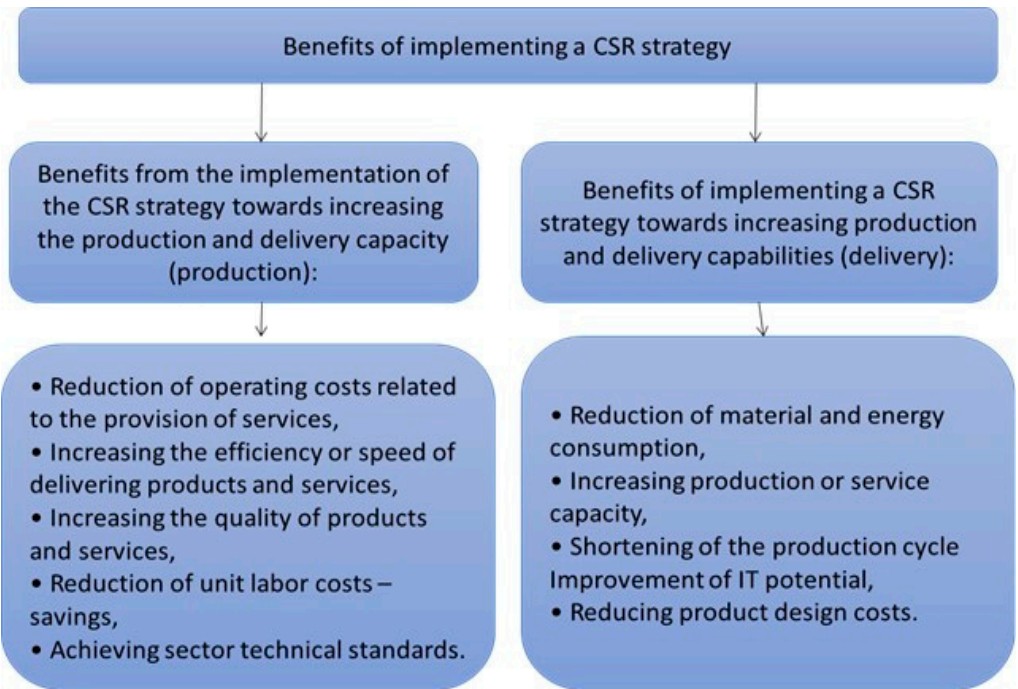

**Figure 8.** Benefits of implementing a CSR strategy towards increasing production and delivery capabilities. Source: own study.

Overall, the discussed model shows that the implementation of CSR strategies in the area of production and supply brings numerous benefits, such as cost reduction, quality improvement, increased efficiency, and environmental protection. These activities contribute to increasing the company's competitiveness on the market and have a positive impact on its image and customer confidence.

Figure 9 shows the benefits of implementing the CSR strategy in the context of better organization of the workplace. This model shows the positive effects that a company can achieve by taking actions in line with CSR principles in the area of workplace organization and management.

**Figure 9.** Benefits of implementing a CSR strategy towards a better organization of the workplace. Source: Authors' study based on the obtained results.

Benefits of implementing a CSR strategy towards better organization of the workplace:

- Strengthening relations with clients: Engaging in CSR activities, such as taking pro-social actions or caring for the wellbeing of employees, can contribute to strengthening relations with clients who appreciate the company's responsible approach.
- Increasing the ability to adapt to different customer requirements: CSR activities, such as taking into account customer expectations regarding social responsibility, can help the company adapt to changing market requirements.
- Improving communication and interaction between different departments in the organization: Implementing a CSR strategy can foster better understanding and cooperation between different departments in the company, which translates into more effective management and decision-making.
- Reducing product design costs: CSR activities, such as ensuring efficient use of resources or reducing waste, can contribute to reducing the cost of designing new products.
- Increased employee motivation: Employees who see the company's commitment to social responsibility are often more engaged and motivated to work.
- Simplification of processes: Implementing a CSR strategy can help simplify processes in an organization by eliminating redundant bureaucratic procedures.
- Better organization: CSR activities are conducive to better organization of the company, both internally and in contact with the environment.

- Improvement of working conditions: Companies committed to CSR often invest in improving the working conditions of their employees, which translates into greater satisfaction and productivity.

Overall, the discussed model shows that the implementation of a CSR strategy in the area of workplace organization brings numerous benefits both for employees and for the company itself. These activities have a positive impact on customer relations, process efficiency, employee motivation, and the perception of the company as a responsible entity, which may contribute to increasing the company's competitiveness and positive impact on its image.

Figure 10 presents an analysis of areas indicating problems with the implementation of the CSR strategy in enterprises. As part of this analysis, five key factors were considered that may constitute barriers to the effective implementation of activities in accordance with the principles of CSR. Each of these areas was subject to a detailed assessment, pointing to the potential challenges that companies may encounter on their way to social and environmental responsibility. Potential solutions and recommendations to overcome these difficulties and promote more responsible business practices will be discussed later in the figure. This analysis is aimed at supporting enterprises in their pursuit of sustainable and socially responsible development, bringing benefits to companies, society, and the environment.

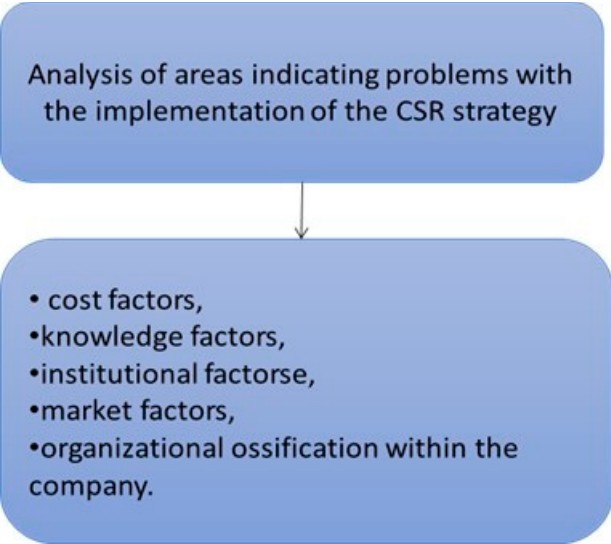

**Figure 10.** Analysis of areas indicating problems with the implementation of the CSR strategy. Source: own study.

The analysis of areas indicating problems with the implementation of the CSR strategy allows us to identify the main obstacles that enterprises may encounter when trying to implement activities in accordance with the principles of CSR. We will discuss each of these areas below:

One of the main problems that companies may encounter when implementing a CSR strategy is the additional costs associated with the implementation of responsible social and environmental activities. Introducing changes in production processes, taking care of sustainable development, or taking prosocial actions may generate higher costs for the company. Companies must find a balance between striving for social responsibility and maintaining a competitive and profitable business.

Insufficient knowledge and awareness of CSR strategies can be another challenge for enterprises. People managing companies may not see the benefits of responsible actions, do not understand the relationship between CSR and business success, or simply do not know how to implement such actions effectively. Education and raising awareness about CSR are crucial in eliminating these types of problems. Companies can also face obstacles related to

regulations and institutional standards. Sometimes unclear laws or a lack of government incentives can make it difficult for companies to implement responsible practices. In some cases, the lack of consistency in enforcing CSR regulations can make companies feel unmotivated to take socially responsible actions.

Market conditions can also influence the implementation of CSR strategies. Competition in the marketplace can make companies focus more on short-term profit goals than on long-term social responsibility. If customers do not appreciate or prefer CSR-compliant products or services, companies may not see incentives to implement them. In some cases, existing organizational structures and company cultures may be resistant to changes related to the implementation of CSR strategies. Organizational ossification, bureaucracy, or employee resistance to new practices may be barriers to the introduction of responsible actions. In such situations, it is important to create an appropriate organizational climate that supports change and encourages CSR activities.

In conclusion, the analysis of the abovementioned areas allows us to understand the potential problems that enterprises may encounter when implementing a CSR strategy. Eliminating these obstacles requires the commitment of both the company's management and all employees, as well as supporting activities at the institutional and market level to create favorable conditions for responsible business practices.

## 5. Discussion

Based on the conducted research, it can be concluded that manufacturing companies in the Silesian Voivodeship show varying degrees of interest and involvement in practices related to corporate social responsibility (CSR). This is observed in small, medium, and large enterprises alike. It was shown that 71% of medium-sized enterprises and 64% of smaller enterprises introduce innovative solutions as part of the CSR strategy. It was also noted that regardless of the company's size, a relatively similar percentage of companies declare the presence of innovative solutions in their CSR strategies.

The analysis shows that the main obstacles to introducing innovations are limited financial resources, outdated technology, and lack of qualified employees. However, no significant correlation was found between the company's age and its innovation capabilities. The results suggest that financial support, technology upgrading, and investing in the development of employee competences can contribute to the increase in enterprise innovation. Regardless of the size of the company, there is significant potential for implementing CSR practices, especially in the context of sustainable development. Knowledge of the CSR concept is relatively common among the surveyed enterprises, which indicates the possibility of further education and awareness-raising activities in this field.

In the context of sustainable development, process innovations have proven to be particularly effective, especially in enterprises with a longer history on the market. It has also been shown that companies with a larger range of operations, i.e., those with a national and international character, face more complex challenges that may require more comprehensive CSR strategies and approaches. Overall, the research results indicate the importance of CSR strategies in the context of increasing the competitiveness and stability of enterprises, as well as in the context of generating positive social and environmental impact. Further research in this area may provide additional insights into effective practices and challenges associated with implementing CSR strategies in various types of enterprises. The analysis of data collected from 310 manufacturing enterprises in the Silesian Voivodeship highlighted several key indicators. Among the surveyed companies, 68% employed from 50 to 249 employees, 24% from 10 to 49 employees, and 8% up to 9 employees. In the context of knowledge of the CSR concept, 74% of respondents from enterprises employing 50–249 people demonstrated understanding of this concept, compared to 69% in companies employing 10–49 people and 54% in companies employing up to 9 people.

In terms of implementing innovative solutions consistent with CSR, 71% of medium-sized enterprises (50–249 employees) and 64% of smaller companies (10–49 employees and up to 9 employees) declared such activities. In research on the types of innovations

consistent with CSR, "completely new" innovations were most often declared by enterprises employing 50–249 people (32%), compared to "improving" (26%) and "adaptive" (22%) innovations. With respect to specific areas of innovation, companies from the category 50–249 employees showed higher rates of innovation in production methods (78%) and in organization (72%) compared to companies in the category of 10–49 employees (31% and 56%, respectively) and companies in the category of up to 9 employees (6% and 34%, respectively). The study provides valuable guidance on diversifying approaches to innovation and CSR in enterprises of different sizes. This may have implications for policies that stimulate innovation and social responsibility in the manufacturing sector. Innovations are crucial for the efficiency and competitiveness of companies. Without innovation, companies may find it difficult to maintain their position in the market and meet the growing needs of consumers.

Innovation is the result of many factors, both internal and external to the company. These factors include the environment in which the company operates, business strategies, willingness to take risks, and the ability to absorb innovation. Scientific research plays a key role in the development of innovation. They enable the identification of innovation conditions and the understanding of innovation processes. However, innovating is not easy and requires both creativity and the ability to turn ideas into practical solutions. Technology plays a key role in innovation, enabling new and improved products to be brought to market and marketing activities to be improved.

Economic development and the introduction of breakthrough innovations are closely related. Many theories, such as Kondratiev's long wave theory or Porter's cluster theory, emphasize the importance of innovation in the economic process. The location where innovation occurs is a key factor influencing economic development. Theories like Porter's cluster theory emphasize the importance of geographical concentration of enterprises and institutions for innovative development. Innovation is not just the result of actions of individuals. Collaboration, thinking together and acting together, as in clusters, can lead to much more innovation. Application of innovation location theory and spatial self-organization theory emphasizes the importance of fluctuation, nonlinearity, and imbalance in the analysis of economic structures. In physics, such states are often the result of differentiation, and in economics, they can lead to differences in the functioning of economic areas, such as cities or regions. The analysis of these fluctuations can therefore help to understand innovation processes. The importance of innovations in social life allows us to conclude that innovations are not limited to the technical and technological sphere. The introduction of new organizational, marketing, or social solutions can be equally innovative and affect the dynamics of companies and organizations. The impact of innovation on a company's competitiveness shows that innovation is crucial to a company's competitiveness, and its ability to innovate can greatly affect its survival and success in the market. A company that is unable to innovate is less likely to survive. The need to create the right environment for innovation includes a work culture that promotes experimentation, rewards success, and tolerates the risk of failure. Organizational structure should also foster innovation by allowing easy access to the necessary resources.

The impact of innovation on customer value allows us to conclude that innovations that provide customers with new value often lead to increased customer loyalty and profitability of the company. Therefore, innovation should always be focused on the needs and benefits of customers. The need to invest in research and development shows what is crucial for the company's innovation and competitiveness. However, in Poland, expenditure on research and development is often insufficient, which may limit the innovative capacity of companies. The key roles of employees and corporate culture indicate that innovation should be promoted, employees should feel safe, and the company's culture should support risky but potentially profitable innovations. Without proper management and corporate culture, innovation can be difficult to achieve. The impact of international trade on innovation shows that international trade can stimulate innovation, improve product quality, and reduce production costs. However, trade barriers can limit companies'

ability to innovate. The impact of business strategies on innovation indicates that choosing the right business strategy is crucial to a company's ability to innovate and compete. A company's organizational structure and strategy should be consistent and support innovation. The dependence of innovation on many factors indicates the ability to invent, implement, and absorb innovations, technical, technological, and financial competences, and structural abilities to strengthen the competitive position.

## 6. Conclusions

The aim of the research was to understand the social and environmental responsibility of innovations undertaken by production companies in the Silesian Voivodeship, with an emphasis on sustainable innovation. The analysis of employment in the surveyed enterprises showed that most of them (68%) are medium-sized companies employing from 50 to 249 employees. This suggests that the manufacturing sector in the study region has a strong representation in the category of medium-sized enterprises. The second-largest group are enterprises that employ from 10 to 49 employees, constituting 24% of the surveyed sample. These are mainly small and medium-sized enterprises. Micro-enterprises and companies employing up to nine employees are the smallest group, representing 8% of the surveyed enterprises. These are primarily small companies with a smaller scope of activity.

Supporting the innovativeness of enterprises towards sustainable innovations can contribute to increasing the competitiveness and stability of the production sector and have a positive impact on the sustainable economic development of the region. For the development of sustainable innovations in enterprises, it is important to provide financial support, investments in modern technologies, and development of employees' competences. The study showed that the main barriers to innovation in the surveyed enterprises are the lack of financial capital, outdated technology, and the lack of qualified employees.

Sustainable innovations, such as those focused on social responsibility and environmental protection, can contribute to increasing the competitiveness of enterprises and sustainable development of the region. The research results provide a new analytical contribution to the understanding of factors influencing enterprise innovation in the context of sustainable development, which can contribute to the development of more effective policies to support innovation in the region. The aim of the research was to understand the social and environmental responsibility undertaken by production companies in the Silesian Voivodeship, with an emphasis on sustainable innovations.

The analysis of the size of employment in the surveyed enterprises showed that most of them are medium-sized companies employing from 50 to 249 employees. Small enterprises (employing up to 9 people) and companies employing from 10 to 49 people constitute a smaller percentage of the surveyed sample. The age of enterprises varies depending on the size of employment. Micro and small enterprises have a similar average age of 19 years, while medium enterprises exist for an average of 24 years.

The age of a company can affect its experience, financial stability, and ability to survive in the market. Older enterprises may have richer experience and a developed network of customers and business partners, which can contribute to their success. Younger companies may be more innovative, but at the same time more exposed to the risks associated with the start-up phase. Enterprises in the researched area have a different scope of activity. The largest group are international companies that operate on a global scale, while a smaller percentage operates only at the regional or national level.

The study showed that most of the surveyed enterprises have some knowledge of the CSR concept, regardless of their size. Larger companies show a slightly higher level of awareness compared to smaller companies. There is potential for CSR education and raising of awareness, especially among smaller businesses, to promote more responsible business practices and generate a positive social and environmental impact.

Enterprises with more employment (50–249 people) more often implement innovative solutions in accordance with CSR assumptions, compared to smaller enterprises. However,

even smaller companies strive to implement such solutions, which may suggest that an innovative approach in line with CSR has potential in different sizes of enterprises and can contribute to sustainable development in various industries and areas of the economy. The majority of medium-sized enterprises (50–249 employees) have implemented innovative solutions in their CSR strategies, with 71% reporting such occurrences. Smaller companies (10–49 employees and up to 9 employees) also show a high level of innovation, with 64% of them having innovative solutions integrated into their CSR strategies. Regardless of the size of the company, a relatively similar percentage of companies declare the presence of innovative solutions in their CSR strategies. Medium-sized companies exhibit a slightly higher percentage (71%), while smaller companies have a slightly lower result (64%), but both groups still prioritize innovation in their CSR activities.

Larger companies (50–249 employees) demonstrate a significantly higher level of innovation across various areas related to CSR, such as innovative production methods (78%), innovation in the organization (activity) (72%), and innovation in marketing (72%). In contrast, smaller companies (10–49 employees and up to 9 employees) generally exhibit lower percentages in these areas, indicating potential areas for improvement. The implementation of innovative solutions in the CSR strategies of all companies, regardless of their size, can lead to increased competitiveness and positive social and environmental impacts. Smaller companies have the potential to enhance their innovation and CSR activities, which can benefit both the company and its stakeholders.

Overall, the findings highlight the importance of innovation in CSR strategies across different-sized enterprises. Companies of all sizes recognize the significance of implementing innovative solutions to enhance their social and environmental responsibilities and remain competitive in the market. However, larger companies generally display a higher level of innovation, indicating the potential for smaller enterprises to further develop their innovative approaches to CSR. Medium-sized enterprises (50–249 employees) exhibit a diverse presence of various types of innovative solutions in accordance with the CSR concept. The highest percentage (32%) of respondents in this group declared the introduction of "completely new" innovations, followed by "aspiring to implement innovation" (25%) and "repeatable innovation" (22%). "Improvement of an existing innovation" was reported by 14% of respondents, and the smallest percentage (7%) belongs to the group declaring a "lack of knowledge" about such innovations. Small enterprises in the category of 10–49 employees also display differences in the occurrence of various types of innovative solutions aligned with CSR. The highest percentage (48%) in this group reported a "lack of knowledge" about such innovations. "Aspiring to implement innovation" accounted for 18%, while "repeatable innovation" and "completely new" innovations constituted 16% and 13%, respectively. The smallest percentage (5%) was attributed to the group focused on the "improvement of an existing innovation". Enterprises in the smallest category, up to nine employees, have the most homogeneous distribution among all categories. The highest percentage (74%) in this group expressed a "lack of knowledge" about innovations consistent with the CSR concept. "Completely new" innovations accounted for 6%, while "repeatable innovation" and "improvement of an existing innovation" constituted 8% each. The smallest percentage (6%) belonged to the group "aspiring to implement innovation".

The results indicate that larger enterprises (50–249 employees) demonstrate greater diversity in the types of innovative solutions aligned with the CSR concept, with a higher percentage of respondents implementing "completely new" innovations and aspiring to implement innovations. In contrast, smaller enterprises, especially those with up to nine employees, tend to show a higher lack of knowledge about innovative solutions compliant with CSR. To address this, educational activities and support for smaller companies may be beneficial in raising awareness about innovation and its positive social and environmental impacts.

In conclusion, the findings highlight the importance of promoting awareness and understanding of innovative solutions aligned with the CSR concept, particularly among smaller enterprises. Encouraging the implementation of innovative practices can lead to

positive social and environmental impacts, benefiting both the companies and the broader community. The implementation of a CSR strategy can significantly impact a company's financial performance, as socially responsible businesses often achieve higher profits. This is primarily due to the positive image they cultivate, leading to greater customer loyalty and increased engagement among employees. As a result, such companies experience improved operational efficiency, ultimately leading to higher profitability. Building a positive image among various stakeholders is one of the key benefits of CSR implementation. By actively engaging in CSR activities, companies can demonstrate their commitment to social and environmental causes, which resonates positively with customers, investors, business partners, and society as a whole. As a consequence, customers tend to prefer companies that actively contribute to the wellbeing of society and the environment, leading to higher levels of trust and brand loyalty.

A well-executed CSR strategy can also lead to enhancements in the quality of products and services offered by a company. By integrating social and environmental responsibility throughout their supply chain, businesses can improve the overall value proposition for customers. This includes factors such as product reliability, safety, and ethical sourcing, all of which contribute to higher customer satisfaction and increased loyalty. Another advantage of adopting a CSR strategy is the increased visibility and exposure of a company's products or services. When a company consistently demonstrates its commitment to social responsibility, it often attracts media attention and garners public interest. As a result, the company's products or services gain broader recognition and become more competitive in the market. CSR implementation also offers several benefits concerning market demand and competitiveness. For instance, socially responsible companies have a higher likelihood of increasing or maintaining their market share. This is because customers are more inclined to choose products or services from companies that demonstrate a genuine concern for social and environmental issues, thus making CSR a crucial factor in influencing consumer decisions. Engaging in CSR activities also opens doors to new markets and customer groups. As the number of socially conscious customers grows, companies that actively embrace CSR principles have a competitive advantage in tapping into these emerging markets and reaching previously untapped customer segments.

CSR strategies often lead to product and service diversification. By prioritizing sustainability and social impact, companies may expand their offerings to include more ecologically responsible and socially beneficial products and services. This not only attracts new customers but also positions the company as an industry leader in terms of responsible business practices. In addition to external benefits, CSR implementation also yields advantages in terms of internal operations. Companies that adopt CSR strategies are likely to reduce operating costs by implementing sustainable resource management and waste reduction practices. This translates into enhanced cost-efficiency and resource optimization within the organization. Implementing CSR can improve the speed and efficiency of product or service delivery. By embracing innovative and optimized production processes, companies can shorten delivery times and enhance overall customer experience, leading to increased customer satisfaction and loyalty. CSR's initiatives go beyond just financial gains and extend to wider social and environmental benefits. By reducing negative impacts on the environment, conserving resources, and investing in employee wellbeing, companies demonstrate their commitment to sustainable business practices, positively influencing their reputation and creating a positive social impact.

In conclusion, embracing a CSR strategy offers a multitude of benefits for companies, ranging from increased financial performance and competitive advantage to enhanced market demand and a positive impact on society and the environment. By prioritizing social and environmental responsibility, companies can establish themselves as ethical and trustworthy entities, thereby creating long-term value and sustainability in the global marketplace. Identifying the main obstacles encountered during the implementation of a CSR strategy is essential to proactively address and tackle these challenges. One significant concern that companies may confront is the potential additional costs associated with

adopting responsible social and environmental activities. Initiatives such as changes in production processes, sustainable development efforts, or prosocial actions might incur higher expenses. Thus, striking a balance between pursuing social responsibility and maintaining a competitive and profitable business becomes a critical consideration for enterprises. Moreover, inadequate knowledge and awareness of CSR strategies poses another hurdle for companies. Company leaders may not fully grasp the benefits of responsible actions, fail to comprehend the link between CSR and business success, or simply lack the know-how to implement such initiatives effectively. Therefore, promoting education and raising awareness about CSR is pivotal in overcoming these obstacles and encouraging a more widespread adoption of responsible practices. Furthermore, regulatory and institutional standards can present challenges for companies attempting to implement CSR initiatives. Unclear laws or the absence of government incentives can hinder the seamless integration of responsible practices. Inconsistent enforcement of CSR regulations may also lead to reduced motivation among companies to undertake socially responsible actions.

Market conditions can significantly influence the implementation of CSR strategies as well. The competitive nature of the marketplace might incline companies to prioritize short-term profit objectives over long-term social responsibility. If customers do not adequately appreciate or prefer CSR-compliant products or services, companies may be less incentivized to invest in such initiatives. In addition to external factors, internal challenges linked to existing organizational structures and company cultures might impede the adoption of CSR strategies. Organizational rigidity, bureaucratic impediments, or resistance from employees towards embracing new practices can become barriers to the successful implementation of responsible actions. In such cases, fostering an appropriate organizational climate that supports change and encourages CSR activities becomes paramount.

In conclusion, the thorough analysis of the areas pinpointing potential problems with the implementation of the CSR strategy highlights the complexity of the task at hand for enterprises. Overcoming these barriers necessitates the commitment of the company's management, active engagement from all employees, and a supportive ecosystem that includes institutional and market-level efforts to foster a conducive environment for responsible business practices. By effectively addressing these challenges, companies can pave the way for sustainable and socially responsible development, benefiting not only their own business but also society and the environment as a whole.

**Author Contributions:** Conceptualization, J.S.-W., R.W., I.M., H.W., R.T. and Ż.N.; methodology, J.S.-W., R.W., I.M., H.W., R.T. and Ż.N.; software, J.S.-W., R.W., I.M., H.W., R.T. and Ż.N.; validation, J.S.-W., R.W., I.M., H.W., M.G.-W., K.S., R.T. and Ż.N.; formal analysis, J.S.-W., R.W., I.M., H.W., R.T. and Ż.N.; investigation, J.S.-W., R.W., I.M., H.W., M.G.-W., K.S., R.T. and Ż.N.; resources, J.S.-W., R.W., I.M., H.W., R.T. and Ż.N.; data curation, J.S.-W., R.W., I.M., H.W., M.G.-W., K.S., R.T. and Ż.N.; writing—original draft preparation, J.S.-W., R.W., I.M., H.W., R.T. and Ż.N.; writing—review and editing, J.S.-W., R.W., I.M., H.W., M.G.-W., K.S., R.T. and Ż.N.; visualization, J.S.-W., R.W., I.M., H.W., M.G.-W., K.S., R.T. and Ż.N.; supervision, J.S.-W., R.W., I.M., H.W., R.T. and Ż.N.; project administration, J.S.-W., R.W., I.M., H.W., M.G.-W., K.S., R.T. and Ż.N.; funding acquisition, J.S.-W., R.W., I.M., H.W., M.G.-W., K.S., R.T. and Ż.N. All authors have read and agreed to the published version of the manuscript.

**Funding:** The research received no external funding.

**Institutional Review Board Statement:** Not applicable.

**Informed Consent Statement:** Not applicable.

**Data Availability Statement:** Data are contained within the article.

**Conflicts of Interest:** The authors declare no conflict of interest.

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
