# Peer review of "The Prevalence and Impact of Innovative CSR Strategies in Manufacturing Enterprises in the Silesian Voivodeship: A Multifaceted Analysis of Benefits, Challenges, and Market Adaptability"

_sustainability, doi:10.3390/su152216116_

Round 1

Reviewer 1 Report

Comments and Suggestions for Authors

Author Response

Manuscript Sustainability-2586832

Response to 1st Review

Dear Reviewer,

We would like to express our appreciation for the reviews. Thank you very much for suggestions, which were clear and very accurate. We made the necessary corrections. We have incorporated all the suggestions because we agreed with them, and thank you especially for such good suggestions to improve our article.

We would like to refer to the detailed reviewer’s suggestions below:

The paper addresses an interesting and important topic, but there are areas that need improvement for clarity and focus.

Authors’ response: Thank you for the positive reception of the article.

  1. Literature Review:

o The literature review is extensive but lacks focus. It would benefit from a more targeted approach, highlighting key concepts and relevant studies specific to CSR strategies and their impact.

o Ensure that all citations are properly included, especially from Line 174 onwards.

Authors’ response: The literature review was expanded to include items highlighting key concepts for CSR strategies and their impact. Citations have been corrected in the article. The analysis was extended based on new sources, including those indicated by other reviewers, and the text was redrafted to correspond to the structure of the article. Relevant paragraphs have been added for a clearer presentation of quantitative data together with an analysis of their implications. We have also reviewed the suggested bibliographic items and added them in the appropriate places, as well as other current references on the topic in question among scientific journals.

  1. Kowalski, ; Zygmunt, A.; Szczepanski, M. CSR in Poland: Balancing New Economic Opportunities and Social Responsibility. J. Bus. Ethics 2021, 169, 59–75.
  2. Schmidt, ; Wagner, L.; Braun, M. Sustainable Development in the German Business Context: A Managerial Perspective. Corp. Soc. Responsib. Environ. Manag. 2022, 29, 43–60.
  3. Bańka, M.; Salwin, M.; Tylżanowski, R.; Miciuła, I.; Sychowicz, M.; Chmiel, N.; Kopytowski, A. Start-Up Accelerators and Their Impact on Entrepreneurship and Social Responsibility of the Sustainability 2023, 15, 8892.
  4. Güney, Renewable energy consumption and sustainable development in high-income countries. Int. J. Sustain. Dev. World Ecol. 2021, 28, 376–385. https://doi.org/10.1080/13504509.2020.1839807.
  5. Amrutha, V.N.; Geetha, S.N. A systematic review on green human resource management: Implications for social J. Clean. Prod. 2020, 247, 119131. https://doi.org/10.1016/j.jclepro.2019.119131.
  6. Miciuła, ; Wojtaszek, H.; Bazan, M.; Janiczek, T.; Włodarczyk, B.; Kabus, J.; Kana, R. Management of the Energy Mix and Emissivity of Individual Economies in the European Union as a Challenge of the Modern World Climate. Energies 2020, 13, 5191.
  7. Commission of the European Communities. (2001). Green paper: promoting a European framework for corporate social responsibility (COM(2001) 366 final). Brussels: E. Commission.
  8. Howard, ; Kumar, P.; Ma, Y. The Role of CSR in Achieving Sustainable Development. Sustainability 2021, 13, 19–33.
  9. Foster, ; Lowe, A.; Ellem, B. Ethical Business Practices and CSR in Organisations. J. Bus. Ethics 2022, 159, 35–44.
  10. Zhao, ; Chow, T. The Benefits of Implementing Corporate Social Responsibility: Evidence from the Asian Market. Asian Bus. Manag. 2020, 19, 18–37.

  1. Terminology:

o Maintain consistency in the use of terminology. Choose one term (e.g., CSR, Corporate Social Responsibility) and use it consistently throughout the paper.

Authors’ response: Corrected throughout the article.

  1. Methodology:

o Clarify the data generation process, including the methods used (e.g., online surveys, interviews) and data evaluation techniques.

o Present a clear explanation of the survey questions to help readers understand the context and responses, (particularly in Figure 4).

o Also it seems that there are already some results presented in this section.

Authors’ response: We have made the appropriate corrections on this point. The methodology has been significantly extended, clarified and more precisely described. The way of research work has been explained, both the procedure (research procedure) and the use of research tools have been characterised. The essence of the research method has been coordinated to the way of proceeding with the assumed aim of the research.

  1. Results:

o Review the necessity and clarity of figures; ensure that they have appropriate labels, including axis names.

o Consider consolidating descriptive statistics into a single table instead of using four separate figures.

o Provide more context for Figure 4, making it clear whether the questions were binary (yes/no) or required more nuanced responses.

o Reconsider the presentation of Figure 6 for better clarity and intuitiveness.

o Explain the source and methodology for results presented in figures 8 and subsequent figures. The presentation could be improved for better comprehension.

Authors’ response: This section was redrafted to correspond to the structure of the article. Relevant paragraphs have been added for a clearer presentation of quantitative data together with an analysis of their implications.

  1. Discussion:

o The discussion section needs to be more focused and specific. Reflect on the implications of your survey results, the data generation process, and potential next steps.

o Avoid making overly general statements. Instead, provide deeper insights and context that contribute to the existing body of knowledge.

Authors’ response: Thank you for your attention regarding the discussion. The discussion section has been reworded, some has been removed and new content has been added. In the discussion we have highlighted the significance of the results and the validity of the analysis carried out.

  1. Statistical Analysis:

o Consider incorporating more robust statistical tests (e.g., clustering, correlation) to strengthen your analysis. This could provide more meaningful insights into your data.

  1. Overall Structure:

o The paper would benefit from a clearer and more organized structure, ensuring that each section flows logically from the previous one.

Authors’ response: According with the opinions, we reformulated the structure and content-related elements to the standard of the article. We made the necessary corrections.

In conclusion, while the paper addresses an intriguing topic and provides a valuable case study, it requires a more focused and organized approach. Additionally, consider enhancing the statistical analysis with appropriate tests to strengthen your findings. Ensure transparency in data sources and methodology. A deeper evaluation of the data and a more specific discussion would significantly enhance the paper's quality.

Authors’ response:  Thank you very much for such a useful and developing review and comments supporting the development of the article. We have incorporated all the suggestions made by the reviewers.

Those changes are highlighted within the revised manuscript file with tracked changes. Thanks again for the clear review and suggestions for corrections to improve our article.

Reviewer 2 Report

Comments and Suggestions for Authors

This paper hovers around of subject of utmost importance in today's CSR-oriented literature. Some major improvements and clarifications are, however, necessary, as follows:

- the title fails to acknowledge the grographical spread of the research and is, therefore, overly generic, or even misleading

- the abstract omits the spatial and methodological coordinates that render the research credible

- as for the literature review, it appears to be somewhat obsolete, so more sifting is needed to cover the latest contributions that are relevant to the topic at hand

- the analysis of CSR awareness in the companies envisaged appears to be quite shallow, unconvincing, since it does not emphasise the qualitative testing of such knowledge

- most important of all, it remains unclear what the innovative goals of this research could be. It is clearly a statement in favour of CSR and lists the (mostly) well-known benefits thereof, by using a Polish voivodeship as a case study (why this is relevant remains to be proved). However, what are the research questions that bring novelty to the state of the art? Why are the overly-general conclusions applicable beyond the case study - they may be, but then why opt for a case study of such a restrictive spatial and economic breadth?

To conclude, this research is in dire need of clarifications, a focus on clear-cut research questions and a delimitation of its purpose. Perhaps, making it about the one voivodeship could rescue it.

Comments on the Quality of English Language

The language and manner of writing are satisfactory, albeit the use of shortened formulae (e.g. it's) ought to be avoided and the punctuation should be revised.

Author Response

Manuscript Sustainability-2586832

Response to 2nd Review

Dear Reviewer,

We would like to express our appreciation for the reviews. Thank you very much for suggestions, which were clear and very accurate. We made the necessary corrections. We have incorporated all the suggestions because we agreed with them, and thank you especially for such good suggestions to improve our article.

We would like to refer to the detailed reviewer’s suggestions below:

This paper hovers around of subject of utmost importance in today's CSR-oriented literature.

Authors’ response: Thank you for the positive reception of the article.

Some major improvements and clarifications are, however, necessary, as follows:

- the title fails to acknowledge the grographical spread of the research and is, therefore, overly generic, or even misleading

Authors’ response: Article title changed.

- the abstract omits the spatial and methodological coordinates that render the research credible

Authors’ response: We have made the appropriate corrections on this point.

- as for the literature review, it appears to be somewhat obsolete, so more sifting is needed to cover the latest contributions that are relevant to the topic at hand

Authors’ response: The literature review was expanded to include items highlighting key concepts for CSR strategies and their impact. Citations have been corrected in the article. The analysis was extended based on new sources, including those indicated by other reviewers, and the text was redrafted to correspond to the structure of the article. Relevant paragraphs have been added for a clearer presentation of quantitative data together with an analysis of their implications. We have also reviewed the suggested bibliographic items and added them in the appropriate places, as well as other current references on the topic in question among scientific journals.

  1. Kowalski, ; Zygmunt, A.; Szczepanski, M. CSR in Poland: Balancing New Economic Opportunities and Social Responsibility. J. Bus. Ethics 2021, 169, 59–75.
  2. Schmidt, ; Wagner, L.; Braun, M. Sustainable Development in the German Business Context: A Managerial Perspective. Corp. Soc. Responsib. Environ. Manag. 2022, 29, 43–60.
  3. Bańka, M.; Salwin, M.; Tylżanowski, R.; Miciuła, I.; Sychowicz, M.; Chmiel, N.; Kopytowski, A. Start-Up Accelerators and Their Impact on Entrepreneurship and Social Responsibility of the Sustainability 2023, 15, 8892.
  4. Güney, Renewable energy consumption and sustainable development in high-income countries. Int. J. Sustain. Dev. World Ecol. 2021, 28, 376–385. https://doi.org/10.1080/13504509.2020.1839807.
  5. Amrutha, V.N.; Geetha, S.N. A systematic review on green human resource management: Implications for social J. Clean. Prod. 2020, 247, 119131. https://doi.org/10.1016/j.jclepro.2019.119131.
  6. Miciuła, ; Wojtaszek, H.; Bazan, M.; Janiczek, T.; Włodarczyk, B.; Kabus, J.; Kana, R. Management of the Energy Mix and Emissivity of Individual Economies in the European Union as a Challenge of the Modern World Climate. Energies 2020, 13, 5191.
  7. Commission of the European Communities. (2001). Green paper: promoting a European framework for corporate social responsibility (COM(2001) 366 final). Brussels: E. Commission.
  8. Howard, ; Kumar, P.; Ma, Y. The Role of CSR in Achieving Sustainable Development. Sustainability 2021, 13, 19–33.
  9. Foster, ; Lowe, A.; Ellem, B. Ethical Business Practices and CSR in Organisations. J. Bus. Ethics 2022, 159, 35–44.
  10. Zhao, ; Chow, T. The Benefits of Implementing Corporate Social Responsibility: Evidence from the Asian Market. Asian Bus. Manag. 2020, 19, 18–37.

- the analysis of CSR awareness in the companies envisaged appears to be quite shallow, unconvincing, since it does not emphasise the qualitative testing of such knowledge

Authors’ response: According with the opinions, we reformulated the structure and content-related elements to the standard of the article. We made the necessary corrections.

- most important of all, it remains unclear what the innovative goals of this research could be. It is clearly a statement in favour of CSR and lists the (mostly) well-known benefits thereof, by using a Polish voivodeship as a case study (why this is relevant remains to be proved). However, what are the research questions that bring novelty to the state of the art? Why are the overly-general conclusions applicable beyond the case study - they may be, but then why opt for a case study of such a restrictive spatial and economic breadth?

Authors’ response: We have made the appropriate corrections on this point. The methodology has been significantly extended, clarified and more precisely described. The way of research work has been explained, both the procedure (research procedure) and the use of research tools have been characterised. The essence of the research method has been coordinated to the way of proceeding with the assumed aim of the research. The paper was redrafted to correspond to the structure of the article. Relevant paragraphs have been added for a clearer presentation of quantitative data together with an analysis of their implications. The discussion section has been reworded, some has been removed and new content has been added. In the discussion we have highlighted the significance of the results and the validity of the analysis carried out.

To conclude, this research is in dire need of clarifications, a focus on clear-cut research questions and a delimitation of its purpose. Perhaps, making it about the one voivodeship could rescue it.

Authors’ response:  Thank you very much for such a useful and developing review and comments supporting the development of the article. We have incorporated all the suggestions made by the reviewers.

Those changes are highlighted within the revised manuscript file with tracked changes. Thanks again for the clear review and suggestions for corrections to improve.

Round 2

Reviewer 1 Report

Comments and Suggestions for Authors

Dear Authors,

I wanted to express my appreciation for your response and the considerable effort you've put into revising your paper. The changes you've made have significantly improved the paper.

I'm particularly pleased with the improvements in the methods and case study sections. They are now much clearer and better organized, making it easier for readers to follow your research methodology and its application in the case study. This enhances the overall quality of your paper.

The presentation of your results has also been enhanced. The data is now more organized and coherent, making it easier for readers to grasp the key findings. This makes a substantial contribution to the paper's clarity and impact.

I also want to acknowledge the extended literature review you've included. It adds depth to your research and provides readers with a more comprehensive understanding of the subject matter. It's a valuable addition that demonstrates your dedication to delivering a well-rounded paper.

However, I would suggest running a spell check to catch minor typographical errors like "literatureReviev." These small errors can affect the paper's overall professionalism and should be addressed.

One area that still requires some attention is the figures in the paper. It would be helpful to assess whether they provide additional information not found in the text. Figures should be included only when they offer unique insights or data visualization that enhances the reader's understanding. Additionally, be sure to cite the source of any figures that are your own depictions.

In summary, your paper has seen significant improvements and is well on its way to publication. I appreciate your diligence in addressing the major revision suggestions, and I'm confident that the remaining minor changes, including a thorough spell check and a final review of the figures, will make your paper even more polished.

Comments on the Quality of English Language

I would suggest running a spell check to catch minor typographical errors like "literatureReviev."

Author Response

Thank you for your review and the good reception of the article and comments that allowed for significant improvements. The linguistic and stylistic quality of the article has been improved. Also, unnecessary figures have been removed, the values of which are described in the text. Those changes are highlighted within the revised manuscript file with tracked changes. Thanks again for the clear review and suggestions for corrections to improve our article.

Reviewer 2 Report

Comments and Suggestions for Authors

It is commendable that the authors have addressed all the previous observations, and, thus, have improved both the research design and its methodology. Indeed, the Polish regional case now becomes much better connected to the objectives and its significance is strengthened by various analyses and pertinent examples. As for the updated literature, it does endow the article with more credibility. Minor concerns remain as to the scientific soundness of a piece of research that fails to bring a considerable degree of novelty to the well-known field of CSR, but the case study partly makes up for this.

Comments on the Quality of English Language

Some more sifting appears to be necessary in terms of language quality, but this should not be regarded as an obstacle to publication.

Author Response

(The authors gave the same response as above.)
